# Relationship between Maternal Vitamin D Levels and Adverse Outcomes

**DOI:** 10.3390/nu14204230

**Published:** 2022-10-11

**Authors:** Heng Zhang, Shumin Wang, Lingjin Tuo, Qixiao Zhai, Jingjing Cui, Daozhen Chen, Dexiang Xu

**Affiliations:** 1Department of Child Health Care, Wuxi Maternity and Child Health Care Hospital, Wuxi 214002, China; 2Department of Toxicology, Anhui Medical University, Hefei 230032, China; 3Wuxi School of Medicine, Jiangnan University, Wuxi 214002, China; 4School of Food Science and Technology, Jiangnan University, Wuxi 214122, China; 5Department of Clinical Laboratory, Wuxi Maternity and Child Health Care Hospital, Wuxi 214002, China; 6Department of Laboratory, Haidong Second People’s Hospital, Haidong 810699, China

**Keywords:** vitamin D, pregnancy, adverse outcomes, mechanisms

## Abstract

Vitamin D (VD), a fat-soluble vitamin, has a variety of functions that are important for growth and development, including regulation of cell differentiation and apoptosis, immune system development, and brain development. As such, VD status during pregnancy is critical for maternal health, fetal skeletal growth, and optimal pregnancy outcomes. Studies have confirmed that adverse pregnancy outcomes, such as preeclampsia, low birth weight, neonatal hypocalcemia, poor postnatal growth, skeletal fragility, and increased incidence of autoimmune diseases, can be associated with low VD levels during pregnancy and infancy. Thus, there is growing interest in the role of VD during pregnancy. This review summarizes the potential adverse health outcomes of maternal VD status during pregnancy for both mother and offspring (gestational diabetes mellitus, hypertensive gestational hypertension, intrauterine growth restriction, miscarriage, stillbirth, and preterm birth) and discusses the underlying mechanisms (regulation of cytokine pathways, immune system processing, internal secretion, placental function, etc.) of VD in regulating each of the outcomes. This review aims to provide a basis for public health intervention strategies to reduce the incidence of adverse pregnancies.

## 1. Introduction

Vitamin D (VD) is a member of the steroid hormone family that includes both VD2 (ergocalciferol) and VD3 (cholecalciferol) forms, but both are biologically inactive in the human body. Vitamin D requires VD-25-hydroxylase action to form 25-hydroxyvitamin D (calcidiol, 25OHD), which needs further activation by a second hydroxylation step catalyzed by the enzyme 25OHD-1-α-hydroxylase to generate 1,25(OH)2D (calcitriol) [1]. It is understood that 1,25(OH)2D is the main component responsible for the biologically active effect of VD in the body, which increases intestinal absorption of calcium and bone resorption and decreases renal excretion of calcium and phosphate [2]. However, 25OHD is the best indicator of VD nutritional status because of its stability and long half-life in the body [3].

In people, the main source of VD is through the action of ultraviolet B radiation on 7-dehydrocholesterol in the skin, with small amounts derived from dietary sources [4]. The vast majority (85 to 90%) of this VD is bound to VD-binding protein (DBP) and stored in the body. A small amount (10% to 15%) is also bound to albumin, with an additional 1% of the total amount of free VD [5]. The diagnostic cutoff points for VD status (deficiency, insufficiency, and adequacy) are not fully harmonized due to several factors, such as latitude, time spent outdoors, ethnicity, and VD supplementation. The Institute of Medicine (IOM) defined VD deficiency as 25OHD concentrations < 20 ng/mL (50 nmol/L), VD insufficiency as 20 to 30 ng/mL (50–75 nmol/L), and VD adequacy as >30 ng/mL (75 nmol/L) in the serum [6].

A high prevalence of VD deficiency or insufficiency has been observed in many populations worldwide. Due to the important role of VD in fetal growth and development, the supply of VD in the pregnancy period needs to cover the demand [7]. Although pregnant women in most countries are encouraged to take daily prenatal vitamin supplements containing VD, the incidence of VD deficiency is disturbingly high among pregnant women (Table 1). Studies have shown that the prevalence of VD deficiency in pregnant women ranges from approximately 26% to 98%, and the prevalence of VD insufficiency is >66% in various countries worldwide. Although there is some variation in the reported prevalence of VD deficiency or insufficiency worldwide, the prevalence of VD deficiency or insufficiency remains high in pregnant women. Many factors affect the VD status of pregnant women, including latitude, season, diet, dietary supplements, time spent outdoors, clothing habits, sunscreen use, weight status, skin color, medications, and medical conditions.

Low maternal VD levels during pregnancy are associated with various adverse obstetric outcomes, such as gestational diabetes mellitus (GDM) [22], preeclampsia [23], and primary cesarean section [24]. Additionally, gestational VD deficiency has been linked to fetal intrauterine growth restriction and multiple adverse fetal and neonatal health outcomes, including a higher risk of preterm birth (PTB) [25], abortion [26], low birth weight [27], and neonatal hypocalcemia [28].

Given the high prevalence of VD deficiency in pregnant women, there is an urgent need to determine the impact of maternal VD status during pregnancy on potential adverse health outcomes in mothers and offspring to design effective prevention strategies that might reverse these worrisome trends. Therefore, this review aimed to determine the relationship between gestational VD status and potential adverse health outcomes and to identify the potential mechanisms by which VD modulates these outcomes (Figure 1).

## 2. Maternal Vitamin D Status and Adverse Pregnancy Outcomes

### 2.1. Gestational Diabetes Mellitus (GDM)

Gestational diabetes mellitus (GDM) is a disease caused by disorders of the glucose metabolism during pregnancy that may increase morbidity and mortality in mothers and neonates, including hypertension, preeclampsia, urinary tract infection, cesarean delivery, fetal macrosomia, neonatal hypoglycemia, and a higher long-term risk for metabolic syndrome or diabetes mellitus type 2 development [29,30]. Low VD status in pregnant women could cause an increased risk of GDM, which may be due to the connection between VD and insulin or glucose metabolism. Furthermore, VD insufficiency may reduce insulin sensitivity by affecting insulin receptor expression and insulin response to glucose. In addition, VD deficiency may contribute to the development of GDM by a 1,25(OH)D2-VDR binding in pancreatic β-cells to break the balance between extra- and intracellular levels [31].

The association between VD and GDM has attracted considerable attention in recent years. A meta-analysis that included 16,515 individuals from 20 observational studies on the correlation between VD status and GDM in a broad range of populations revealed that VD deficiency significantly increased the risk of GDM by 45% [32]. In another prospective birth cohort study, Yin et al. [33] followed 4984 pregnant women and found that the GDM risk was significantly lower in women with 25OHD concentrations ranging from 50 to 75 nmol/L and >75 nmol/L than in women with 25OHD concentrations of < 25 nmol/L. The curve-fitting models suggested a significant reduction in GDM risk, fasting plasma glucose, and area under the curve of glucose with increasing 25OHD concentrations only for concentrations > 50 nmol/L. Furthermore, in a case-control study, GDM pregnant women (24–28 weeks of gestation) with VD deficiency (<50 nmol/L) treated with VD3 1200 IU/d had a significant increase in 25OHD serum levels and a significant decrease in fasting plasma glucose, postprandial 2 h plasma glucose, and glycosylated hemoglobin at 36 weeks of gestation, which supports the positive glucose metabolic effects of VD3 supplementation on mothers [34]. Regarding animal models, the risk of GDM was found to be increased in guinea pigs with low VD status during pregnancy. Although intervention by dietary VD intake during pregnancy did not affect the occurrence of GDM, a higher pre-pregnancy VD status appears to be protective [35]. 

The pancreas contains both cytochrome P450c27B enzymes for the local production of 1,25(OH)2D and the VD receptor [36]. The known physiological mechanism linking 1,25(OH)2D and glucose metabolism is that 1,25(OH)2D regulates intracellular calcium fluxes in β-pancreatic cells and, therefore, regulates depolarization-stimulated insulin release [37]. However, the mechanisms by which hypovitaminosis D (VD insufficiency and deficiency) influences GDM may extend beyond insulin metabolism to the genes with the VD response element [38]. It has been suggested that the relationship between GDM and VD status may be mediated by a single nucleotide polymorphism in the *CYP27B1* (1-α-hydroxylase) promoter region [39]. These findings underscore the need for VD supplementation trials for women at high risk for GDM.

### 2.2. Pregnancy-Induced Hypertension

Pregnancy-induced hypertension (PIH) is a group of disorders with abnormal regulation of blood pressure during pregnancy, including gestational hypertension, preeclampsia, eclampsia, chronic hypertension with preeclampsia, and chronic hypertension [40], which are risk factors for maternal and perinatal mortality [41].

Numerous studies have found that maternal VD status during pregnancy is strongly associated with PIH [42,43,44]. Furthermore, PIH may be associated with abnormal local synthesis of active 1,25(OH)2D from the precursor 25OHD, and low levels of maternal 25OHD may further impair the production of 1,25(OH)2D in the placenta [45]. Additionally, VD may also affect PIH by modulating the renin–angiotensin–aldosterone system, a regulatory cascade that plays a key role in the regulation of blood pressure, electrolyte balance, and body fluid homeostasis [46,47]. A prospective cohort study found that serum 25OHD concentration was significantly lower in patients with preeclampsia compared with normal pregnant women (118 nmol/L ± 44.25 vs. 130.75 nmol/L ± 4.3, *p* < 0.01) and that the risk of preeclampsia was 2.48-fold higher in the hypovitaminosis D group compared with the VD adequate group [95% confidence interval(CI): 1.51–4.08)] [48]. It is worth mentioning that the results remained significantly different after excluding the effects of pre-pregnancy body mass index (BMI), maternal age, smoking, the number of deliveries, season of blood collection, week of gestation at the time of blood collection, and region of the cohort population. Haugen and colleagues [49] found a 27% reduction in the risk of preeclampsia in women receiving 400–600 international unit (IU) /day of VD from supplements at mid-pregnancy compared with women not receiving supplementation (adjusted odds ratio (OR) = 0.73, 95% CI: 0.58–0.92), strongly suggesting that pregnant women should have VD sufficiency to lower the risk of preeclampsia development. Similarly, an American study revealed that a 25 nmol/L increase in 25OHD levels yields a 63% decrease in the risk of severe preeclampsia [50]. Independent meta-analyses have reported that the risk of preeclampsia was significantly increased in women with VD insufficiency or deficiency compared with control groups [51,52,53] and that VD was the only metabolite in common for preeclampsia and gestational hypertension prediction among the 122 different metabolites [54]. 

Similar results have also been reported in animal studies. Liu and colleagues [55] studied a model of pregnant VD-deficient BL6 mice and concluded that both systolic blood pressure and mean arterial pressure were significantly increased on day 14 of gestation, and that the vascular diameter of the placental labyrinth region was reduced compared with the VD-normal group. A preeclampsia rat model induced by reduced uterine perfusion pressure (RUPP) showed that VD supplementation reduced the pathophysiology and hypertension associated with preeclampsia [56]. Specifically, VD treatment reduced CD4 T cells, angiotensin type 1 receptors, endothelin-1, soluble FMS-like tyrosine kinase-1, and blood pressure in the RUPP rat model of preeclampsia and, thus, VD supplementation could be an avenue to improve the treatment of hypertension in response to placental ischemia.

Contrary to these findings, some studies have concluded that there is no significant correlation between VD status and preeclampsia. Al-Shaikh et al. [57] conducted a cross-sectional study on the association between VD and birth outcomes in 1000 pregnant Saudi women, revealing no significant relationship between low serum 25OHD and PIH. Interestingly, in a large prenatal cohort in America, it was found that higher 25OHD concentrations were associated with higher odds of PIH, as with every 25 nmol/L increase in plasma 25OHD concentration, the risk of developing PIH increased 1.32-fold (95% CI: 1.01–1.72) [58]. Although this finding may be due to chance, the results were somewhat surprising.

Inconsistent results regarding the association between maternal VD status during pregnancy and PIH may be related to multiple confounding factors, such as race [59], season [60], diet (including VD intake and patterns) [61], and the method of measuring 25OHD levels [62,63]. Hence, further multicenter studies with larger sample sizes are needed to determine the serum levels and degree of supplementation required to optimize maternal outcomes. Finally, most studies have confirmed that VD levels are deficient in patients with PIH, and there is an increased prevalence of VD deficiency with PIH, suggesting that VD deficiency can be a risk factor for the development of PIH.

### 2.3. Spontaneous Abortion and Stillbirth

Spontaneous abortion and stillbirth are serious adverse outcomes in pregnancy and can cause psychological distress in a subsequent pregnancy in women [64]. A Chinese cross-sectional study revealed that low VD levels in pregnant women increased the risk of spontaneous abortion by 1.71-fold (95% CI: 1.2–2.4) by measuring serum 25OHD concentrations from 60 nulliparous women with singletons in early gestation (7–9 weeks) [65]. Barebring et al. [66] also demonstrated that a higher VD in early pregnancy was associated with a lower risk of spontaneous abortion, with every 1 nmol/L increase in serum 25OHD concentration associated with a 1% reduction in the risk of spontaneous abortion (OR = 0.989, 95% CI: 0.98–1.00; *p* < 0.05). Similarly, Andersen and colleagues [67] conducted a prospective cohort study in Denmark to investigate whether a 25OHD serum concentration was a modifiable risk factor for early spontaneous abortion. The results showed that the adjusted hazard of first-trimester spontaneous abortion was lower in individuals with higher 25OHD concentrations (hazard ratio (HR) = 0.98; 95% CI: 0.96, 0.99), but a 25OHD concentration was not associated with an increased risk of second-trimester spontaneous abortion. In light of these findings, it is even more evident that VD shortage has a vital influence on pregnancy consequences.

Available evidence suggests that low VD levels are not associated with stillbirths. In a prospective observational study, maternal 25OHD levels were measured in 2960 pregnant women at 16–20-week gestation; 18.9% and 48.6% of pregnant women had low and medium levels of VD, respectively [68]. However, there was no significant difference in stillbirths among the pregnant women with different VD levels. In another nested case-control study, Schneuer et al. [69] measured serum 25OHD levels in 5109 women at 10–14 weeks in Australia and assessed its association with adverse pregnancy outcomes via multivariate logistic regression. After adjusting for maternal and clinical risk factors, they confirmed that low 25OHD serum concentrations in the first trimester of pregnancy were not associated with adverse pregnancy outcomes, including small for gestational age (SGA), PTB, preeclampsia, GDM, miscarriage, and stillbirth.

Evidence regarding VD nutrition and its relationship with all adverse outcomes is controversial. The risk of spontaneous abortion or stillbirth may be associated with multiple complicated factors, including consanguineous marriage [70], age [70], BMI [71], history of chronic disease [72], passive smoking [73], and alcohol intake [71]. Therefore, the benefits of VD supplementation during pregnancy should be evaluated further through rigorous interventional studies.

### 2.4. Preterm Birth (PTB)

Preterm birth (PTB), defined as delivery before 37 weeks of gestation, is an important risk factor for neonatal mortality, morbidity, and developmental abnormalities during childhood [74]. A prospective cohort study on 2327 pregnant women showed that low maternal serum 25OHD concentration (<50 nmol/L) significantly increased the incidence of PTB (<37 weeks), and the results were similar when limited to cases that were medically indicated or occurred spontaneously and cases occurring at <34 weeks of gestation [25]. Moreover, the risk of PTB (<37 weeks) significantly decreased when serum 25OHD concentration reached approximately 90 nmol/L. Furthermore, Perez-Ferre et al. [75] evaluated the association between maternal serum 25OHD status and newborn outcomes in Spanish women. They showed that VD deficiency was prevalent during pregnancy (59%, second trimester), and lower 25OHD levels were associated with PTB. The cutoff with the best combination of sensitivity and specificity was 35 nmol/L (66.7% sensitivity and 71.0% specificity). These findings support a protective role of maternal VD sufficiency in PTB, which may provide justification for a randomized clinical trial of maternal VD replacement or supplementation to prevent PTB. Consistently, in a mouse model of VD and calcium diet restriction during pregnancy, VD deficiency caused abnormalities in placental morphogenesis and fetal growth, indicating that an interactive effect of low calcium and VD intake during pregnancy may also increase the PTB rate [76]. 

The protective effect of maternal VD during pregnancy on the development of PTB remains controversial. Prospective cohort studies by Shand et al. [48] and Yang et al. [77] revealed no significant difference in the incidence of PTB among pregnant women with different serum 25OHD concentrations. Of particular interest is the prospective observational study conducted by Zhou and colleagues [68]. In this study, pregnant women (*n* = 2960) and healthy controls (*n* = 100) were recruited to measure maternal 25OHD levels at 16–20 weeks of gestation. Interestingly, women with high levels of VD (≥75 nmol/L) had a higher incidence of PTB than those in the 25OHD deficiency (<50 nmol/L) and insufficient (50–75 nmol/L) groups, although the authors emphasized that this could be related to the older age. Possible risk factors for PTB include an unhealthy lifestyle, mental stress, younger or older age during pregnancy, and malnutrition [78,79]. In addition, thyroid function showed a close relationship with PTB [80,81]. Such confounding factors make it difficult to directly compare the result of different studies, which might account for the inconsistencies. 

### 2.5. Intrauterine Growth Restriction

Intrauterine growth restriction (IUGR) is a common and complex obstetric problem, defined as a fetus failing to achieve growth potential, which clinically manifests as low birth weight (LBW) or SGA infants [82,83,84]. The frequency of IUGR has been reported to be approximately 10–15% [84]. As such, IUGR is a major public health concern worldwide and is associated with high perinatal morbidity and mortality [84]. Moreover, infants born with IUGR have been reported to have an increased incidence and prevalence of many diseases, including decreased intelligence and cognition [85], short stature [86], insulin resistance [87], and chronic lung disease [88]. 

Miliku et al. [89] performed a regression analysis of 25OHD concentrations in 7098 Dutch pregnant women, whose venous blood samples were collected in the second trimester (18.5–23.3 weeks). Compared with the highest 25OHD quartile group (>P75), a significantly lower quartile (<P25) was associated with offspring having third-trimester fetal growth restriction, leading to a smaller head circumference, shorter body length, and lower body weight at birth. Chen et al. [90] performed a population-based birth cohort study on 3658 mother-and-singleton-offspring pairs to investigate the correlation between maternal VD deficiency during pregnancy and the risk of LBW or SGA in infants. The results showed a positive correlation between maternal serum 25OHD levels and offspring birth weight (r = 0.477; *p* < 0.001). 

After adjusting for confounders, the incidence rate of LBW at birth was 12.31% (95% CI: 4.47, 33.89) among subjects with VD deficiency, and 3.15% (95% CI: 1.06, 9.39) among subjects with VD insufficiency. The incidence rate of SGA infants was 6.47% (95% CI: 4.30, 9.75) and 2.01% (95% CI: 1.28, 3.16) among subjects with VD deficiency and insufficiency, respectively. The results are consistent with data derived from a large multi-ethnic cohort from the Netherlands (Amsterdam Birth Children and their Development cohort) that included 3730 women with live singleton full-term births [91] and from an American prospective prenatal cohort study that included 1067 white and 236 black mother–infant pairs [92]. Further analysis found that gestational VD deficiency may cause placental insufficiency and fetal IUGR, partially by inducing placental inflammation [93].

The maternal 25OHD exposure window during pregnancy may be important for fetal growth in utero. The interaction between VD and many other hormones and nutrients affects fetal growth [94]. For instance, both maternal calcium absorption and placental calcium transfer are increased during pregnancy to meet fetal demands and in response to 25OHD [95]. Calcium is a key structural component of bone development, and a higher concentration of calcium is required to effectively mineralize fetal bone [80]. Therefore, the role of VD in calcium absorption may also affect fetal skeletal muscle and bone development.

## 3. Potential Mechanisms of Maternal VD Status during Pregnancy and Adverse Pregnancy

During pregnancy, maternal calcium mobilization increases, and a number of physiological adaptations occur, including increased maternal serum 1,25(OH)2D, DBP, placental VDR, and renal and placental CYP27B1 activity to maintain normal serum 25OHD and calcium levels [96]. Maternal 25OHD crosses the placenta and is the main form of VD in the fetus. The mechanism of maternal VD deficiency during pregnancy has not been fully elucidated. However, recent studies have found that affected pathways may be associated with several factors, such as oxidative stress, imbalance in the regulation of the inflammatory response, and compromise of placental function during pregnancy [97,98].

### 3.1. Regulation of Cytokine Pathways

Studies have confirmed that VD deficiency may increase placental inflammation, impair placental function, and cause adverse pregnancy outcomes [99]. Vitamin D is involved in biological processes, such as immunity and inflammation, via binding to VDR, which is a member of the superfamily of nuclear receptors [100]. Maternal VD during pregnancy activates placental trophoblast VDR, which binds to nuclear factor kappa-B (NF-κB) and blocks NF-κB nuclear translocation, thereby downregulating peroxisome proliferator-activated receptor γ (PPARγ) and reducing inflammatory factor levels [101]. Conversely, VD deficiency suppresses VDR expression, thereby increasing PPARγ transcriptional activity and inflammation levels. Additionally, VD regulates the immune system and inhibits inflammation by inhibiting inflammatory cytokines, including tumor necrosis factor-α (TNF-α), interferon-γ (IFN-γ), and interleukin 6 (IL-6) [30]. These results suggest that VD can regulate the cytokine pathways, and placental inflammation during pregnancy may be related to VD deficiency.

The role of VD in the regulation of maternal inflammation was also demonstrated in a mouse model of bacterial lipopolysaccharide (LPS)-induced adverse pregnancy outcomes [102]. Indeed, LPS can activate NF-κB via Myeloid differentiation factor 88 (MyD88)-dependent and Toll/IL-1R-domain-containing adaptor protein inducing IFN-β(TRIF)-dependent pathways via the pattern recognition Toll-like receptor (TLR) [103]. Pro-inflammatory cytokines (IL-8, TNF-α, and IL-6) are then secreted, which leads to stillbirth, PTB, and abortion [104]. Vitamin D supplementation in pregnant mice showed an unexpected ability to counteract LPS-induced secretion of inflammatory cytokines. Zhang et al. [102]. revealed that VD reversed the transcriptional and T helper 17 (Th17) cell differential profiles of offspring CD4^+^ T lymphocytes induced by intrauterine LPS and indicated the contribution of maternal VD supplementation to immune protection in offspring affected by intrauterine inflammation. Vitamin D supplementation decreased lymphocyte differentiation and activation and increased the response to viruses and bacteria in offspring CD4^+^ T cells upon intrauterine LPS exposure. In addition, several pathways, including the T-cell receptor signaling pathway, mitogen-activated protein kinase signaling pathway, Th17 cell differentiation, and autophagy, were downregulated by intrauterine VD intervention following LPS exposure. An earlier study also confirmed that the VD-activating enzyme 1α-hydroxylase (CYP27B1) and VDR support an anti-inflammatory response to VD in the placenta. Liu and colleagues [105] treated wild-type placentas ex vivo with 25OHD_3_, a substrate of Cyp27b1, which inhibited the LPS-induced expression of IL-6 and chemokine Ccl11.

These findings suggest that maternal VD during pregnancy plays a key role in controlling placental inflammation (Figure 2). In humans, maternal VD may be an important factor in the placental response to infection and the associated adverse pregnancy outcomes.

### 3.2. Regulation of Immune System Processing

The VDR and 1,25(OH)2D are present in a variety of female reproductive organs, such as the pituitary glands, hypothalamus, uterus, oviducts, ovaries, mammary glands, and placenta [106]. Vitamin D plays an important role in embryo implantation, placental formation, differentiation, and maturation of trophoblast cells [107]. It has been shown that VD can cross the placental barrier into the fetus and be indispensable for the maintenance of pregnancy in humans as an immunomodulator [108].

It has been demonstrated that 25OHD can be converted to 1,25(OH)2D, active D3, by placental hydroxylase, while both maternal meconium and fetal trophoblast cells show a high activity of 1α-hydroxylase (CYP27B1) [108,109]. The autocrine metabolism of 25OHD to 1,25(OH)2D promotes the immune response in the maternal meconium and placental trophoblast cells [110,111]. This suggests an important role of VD in conception, including implantation and placental development. The immunomodulatory role of VD may be compromised in a low maternal 25OHD state, with potentially damaging effects on placental physiology [112,113].

In vitro and in vivo experiments have shown that dysregulation of placental VD metabolism (*CYP27B1*-knockout) or dysfunction (*VDR*-knockout) stimulates abnormal immune responses [114,115]. In another study, VD injection increased uterine weight and promoted decidualization of the endometrium in pseudo-pregnant rats, suggesting that VD plays a crucial role in blastocyst implantation [116].

After implantation, Tregs maintain maternal tolerance by suppressing cytotoxic T cells, Th1 cells, macrophages, dendritic cells (DCs), and natural killer (NK) cells. Indeed, Tregs are a subpopulation of T lymphocytes that suppress the immune system’s destructive response and protect against autoimmune diseases during pregnancy [117]. Furthermore, 1,25(OH)2D promotes the effector function of Tregs, which have immunosuppressive functions and are essential for the establishment of pregnancy [118,119]. In endometrial stem cells, 1,25(OH)2D reduces the production of most cytokines, such as IL-6, which prevents Treg development but upregulates transforming growth factor-β that activates Tregs. In addition, 1,25(OH)2D promotes DCs with tolerogenic properties by inhibiting their maturation [120], decreases the production of IL-12, which is capable of activating Th1 cells, and increases the production of Th2 cytokine IL-10 in tolerogenic DCs. The DCs also play an important role in Treg activation. The correct balance between Th1 cytokines, such as TNF-α, INF-γ, and IL-2, and Th2 cytokines, such as IL-4, IL-5, IL-6, IL-9, IL-10, and IL-13, is of great importance for a healthy pregnancy [99].

The adaptive immune system regulates maternal immune tolerance to the fetus during pregnancy. The dominance of Th2 cells and humoral immunity is generally associated with normal pregnancy [121], while 1,25(OH)2D has been shown to selectively suppress Th1 cells and enhance Th2 differentiation by directly affecting CD4^+^ progenitor cells [122]. By reducing Th1 cytokines and promoting Th2 cytokines, 1,25(OH)2D makes the maternal immune system particularly sensitive to pathogens while weakening the self-destructive mechanism of effector T cell subsets [122,123]. Ikemoto et al. [124] found that more than 80% of infertile women were VD insufficient or deficient, with nearly half of these having an elevated Th1/Th2 ratio. Interestingly, the Th1/Th2 ratio was significantly reduced upon VD supplementation. In the innate immune system, uterine NK (uNK) cells have also been shown to respond to VD regulation. The uNK cells are involved in the regulation of spiral artery remodeling and trophoblast invasion, which are essential for successful implantation [125], and 1,25(OH)2D induces uNK cell activation. Evans et al. [112] used primary cultures of human decidual cells from pregnancies to demonstrate that decidual NK cells decreased the synthesis of cytokines, such as granulocyte–macrophage colony-stimulating factor 2, TNF-α, and IL-6, after treatment with 1,25(OH)2D and 25OHD for 28 h.

In conclusion, VD supports placental development and function through its immunomodulatory role, which is critical for conception, placentation, pregnancy progression, and pregnancy outcome [119].

### 3.3. Regulation of Internal Secretion

Vitamin D promotes fetal growth and development by regulating calcium homeostasis and thyroid hormone levels [126]. Maternal VD and calcium levels are altered during pregnancy to support fetal calcium homeostasis. Many adaptive mechanisms involve increased intestinal calcium absorption, renal calcium conservation, and changes in bone metabolism [127]. These adaptations are mediated by changes in the secretion of various calciotropic hormones (1,25(OH)2D, parathyroid hormone, and calcitonin) [127]. Vitamin D is directly or indirectly involved in all these adaptive mechanisms.

Vitamin D acts as a regulator of calcium homeostasis and transport, and maternal 1,25(OH)2D may improve poor IUGR outcomes by affecting the development of skeletal muscle and bone [128,129]. Maternal parathyroid hormone levels increase when VD levels are not sufficient to affect bone resorption to maintain adequate maternal serum calcium levels [130]. The association between bone resorption and low VD levels was also reinforced by the negative association of serum 25OHD level < 50 nmol/L with type 1 collagen cross-linked C-terminal telopeptide in pregnant women [131].

Pregnancy may be associated with changes in iodine homeostasis and other physiological changes, ultimately leading to altered thyroid function [132]. Maternal thyroid function should remain normal, especially in the first trimester, when the fetus is fully dependent on maternal thyroid hormones for brain development [133]. Several studies have shown that VD deficiency and hypothyroidism cause a series of adverse outcomes during pregnancy, including gestational hypertension [134], preeclampsia [135], and premature birth [136]. Rostami et al. [137] assessed the relationship between serum VD levels and thyroid hormones in the first trimester of pregnancy in Iran, showing a significant relationship between VD deficiency and thyroxin (T4) levels during early pregnancy. Moreover, hypothyroidism is common in pregnant women with sufficient iodine nutrition, and autoimmune thyroid disease is the most common cause of hypothyroidism [138]. A prevalence case-control study that included 161 cases with Hashimoto’s thyroiditis (HT) and 162 healthy controls demonstrated that the prevalence of VD insufficiency in HT cases (148 out of 161, 92%) was significantly higher than in healthy controls (102 out of 162, 63%, *p* <  0.0001) [139]. Appropriately 25% of pregnant women with subclinical and hidden hypothyroidism are not explicitly diagnosed in high-risk groups [140].

Vascular endothelial growth factor (VEGF) is a potent regulator of placental vascular function [141]. In vitro cellular experiments revealed that 1,25(OH)2D increased the expression and release of VEGF in rat vascular smooth muscle cells [142]. Furthermore, VEGF protein expression was upregulated 1.74-fold after 24 h and 2.47-fold after 4 days of 1,25(OH)2D treatment. The results suggest that VDR activation by VD supplementation upregulates the expression of its downstream target gene *VEGF* and reduces the risk of adverse outcomes, such as gestational hypertension, preeclampsia, and offspring preterm delivery caused by maternal VD deficiency during pregnancy.

There have also been studies on trophoblasts and 1,25(OH)2D to explore other mechanisms related to placental endocrine function. These studies include the stimulation of human placental lactogen synthesis and release [143], human chorionic gonadotropin expression [144], and regulation of estradiol and progesterone synthesis [145].

### 3.4. Regulation of Placental Function

The function of the placenta is mainly in the following aspects: (1) to provide oxygen to the fetus and metabolize carbon dioxide gas produced by the fetus [146], (2) to provide nutrients and secrete growth factors for the fetus [147], and (3) to protect the fetus from the toxic effects of exogenous substances [148]. Hence, the placenta has several functions, such as metabolism, nutrition, and barrier protection, in terms of embryonic development. Impairment of placental function may lead to miscarriage, preterm delivery, and stillbirth [149,150]. 

In rodent models, it was observed that the ratio of placental labyrinth zone area to junctional zone area was significantly decreased in mice fed a calcium- and VD-deficient diet compared with control-fed mice, suggesting disproportionate changes in the placental structure [76]. This means that nutrient exchange within the labyrinthine zone is less impeded, with increased placental efficiency and a sudden increase in fetal growth, which may lead to preterm delivery.

Folate plays a crucial role as a 1-carbon donor required for de novo synthesis of cellular DNA [151]. There is growing evidence that folate deficiency during pregnancy is a major cause of fetal neural tube defects [152]. Chen et al. [153] exposed pregnant rodents to LPS to model maternal infection, showing that, although VD alone had no effect on placental folate transporter protein expression, supplementation with VD during pregnancy significantly attenuated LPS-induced downregulation of placental folate transporter protein, improved placental folate transport from the maternal circulation to the developing embryo, and prevented LPS-induced fetal neural tube defects.

Prenatal overexposure to glucocorticoids can dramatically alter fetal structure and function [154]. Although glucocorticoids are highly lipophilic and readily diffuse across the placenta, fetal glucocorticoid levels remain significantly lower than maternal levels throughout pregnancy, suggesting that the placental barrier protects the fetus from the harmful effects of glucocorticoid overexposure [154]. Tesic et al. [155] demonstrated that maternal VD deficiency decreases placental expression of 11β-hydroxysteroid dehydrogenase type II, which exposes the developing fetus to higher levels of glucocorticoids. The placental and fetal expression of the high glucocorticoid-sensitive factor glucocorticoid-induced leucine zipper correspondingly increased. Early exposure to high levels of glucocorticoids during development has long-term ramifications for future health outcomes in the offspring in terms of cardiometabolic and neuropsychiatric disorders [156].

## 4. Discussion

Currently, the results of epidemiological studies on the association between maternal VD status during pregnancy and adverse pregnancy outcomes are, to an extent, inconsistent. This inconsistency is specifically manifested by controversial results, different gestation periods, inconsistent association strength, etc. We believe that the discrepancy may be explained by the differences in study design (prospective cohort study, case-control study, and clinical randomized controlled study), inclusion and exclusion criteria, sample size, gestational age of the study population, VD status testing methods, VD deficiency definition cutoffs, and definition of adverse birth outcomes, together with the duration of outdoor light exposure, VD supplementation status, diseases affecting VD metabolism, race and genes of participants, and confounding factors. In addition, the mechanisms linking maternal VD status during pregnancy to adverse pregnancy outcomes have not yet been fully elucidated.

Therefore, further population studies with large prospective cohorts and multicenter clinical randomized controlled trials are required. The inclusion and exclusion criteria, VD level assays, and definitions of outcome variables should be standardized in these population studies. Additionally, confounding factors need to be controlled as much as possible to explore the realistic impact of maternal VD status on pregnancy outcomes. The role of VD supplementation interventions during pregnancy (window period, supplementation dose, and regimen) in improving adverse pregnancy effects should also be explored through clinical randomized controlled trials. It is noteworthy to mention whether excessive VD supplementation during pregnancy has harmful effects on the mother and the offspring. To make adequate decisions about VD supplementation, every individual clinical situation must be analyzed and placed in the correct balance of risk and benefit before prescribing VD supplementation.

Regarding mechanistic exploration, rodent models (e.g., mice) of maternal VD deficiency can be constructed by dietary restriction (VD-deficient diet) or specific gene knockout (*Cyp27b1^+/−^*) to observe the effects of maternal VD deficiency on adverse pregnancy outcomes (GDM, PIH, spontaneous abortion, stillbirth, PTB, and IUGR). Meanwhile, the underlying mechanisms of the role of inflammation, immunity, internal secretion, and placental functions in the adverse outcomes mediated by maternal VD deficiency can be analyzed in cellular experiments (e.g., placental trophoblast cells). In addition, by constructing animal models of GDM, PIH, spontaneous abortion, stillbirth, and PTB during pregnancy, it could be observed whether VD supplementation or other interventions (drugs) during pregnancy improve adverse pregnancy status, thus, providing a reference for reducing the occurrence of adverse pregnancy outcomes and achieving early intervention.

## 5. Conclusions

Although findings on the association between maternal VD status and pregnancy outcomes are not entirely consistent, there is growing evidence that VD deficiency during pregnancy increases the risk of several adverse events that could potentially threaten pregnancy, such as GDM, PIH, spontaneous abortion, stillbirth, PTB, and IUGR. Although more interventional and basic studies are needed to understand the role of VD in pregnancy health and disease, through the information reviewed herein, it is clear that many beneficial effects of VD during gestation involve its regulation of cytokine pathways, immune system processing, internal secretion, and placental function. Vitamin D supplementation during pregnancy may be a safe and accessible way to reduce the incidence of adverse events in the mother and infant. In general, adequate sun exposure, a VD-rich diet, and physical activity should always be considered as the first recommendations, while additional VD supplementation may be advised for pregnant women with severe VD deficiency.

## Figures and Tables

**Figure 1 nutrients-14-04230-f001:**
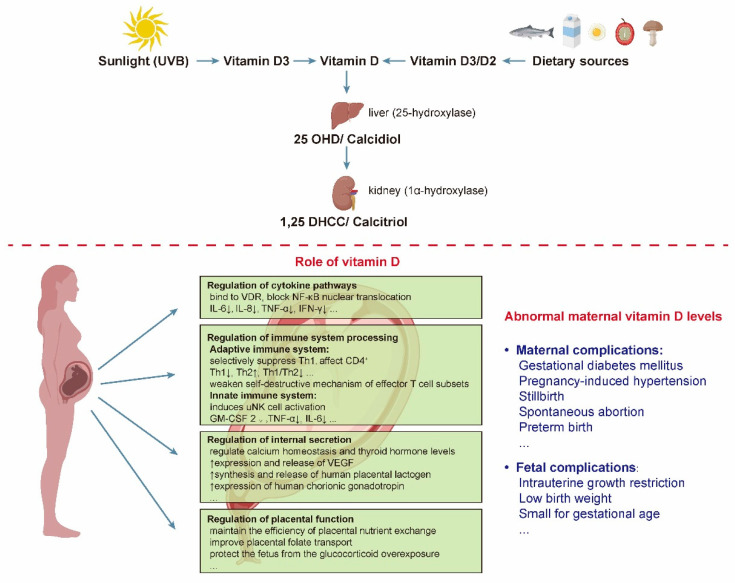
The steps of vitamin D activation and the specific effects of vitamin D on the mother and fetus. There are two forms of vitamin D, namely vitamin D2 and vitamin D3. Vitamin D2 is found in some foods, such as mushrooms, cocoa and chocolate, while vitamin D3 is made by the body on exposure to sunlight or from some foods, such as fish, meat, and fortified dairy products. Abbreviations are as follows: VDR, VD receptor; NF-κB, nuclear factor kappa-B; TNF-α, tumor necrosis factor-α; IFN-γ, interferon-γ; IL, interleukin; Th, T helper; uNK, uterine NK; GM-CSF, granulocyte-macrophage colony-stimulating factor; VEGF, vascular endothelial growth factor; 25 OHD, 25-hydroxyvitamin D; 1,25 DHCC, 1,25-dihydroxychotecalciferol; UVB, ultraviolet-B radiation.

**Figure 2 nutrients-14-04230-f002:**
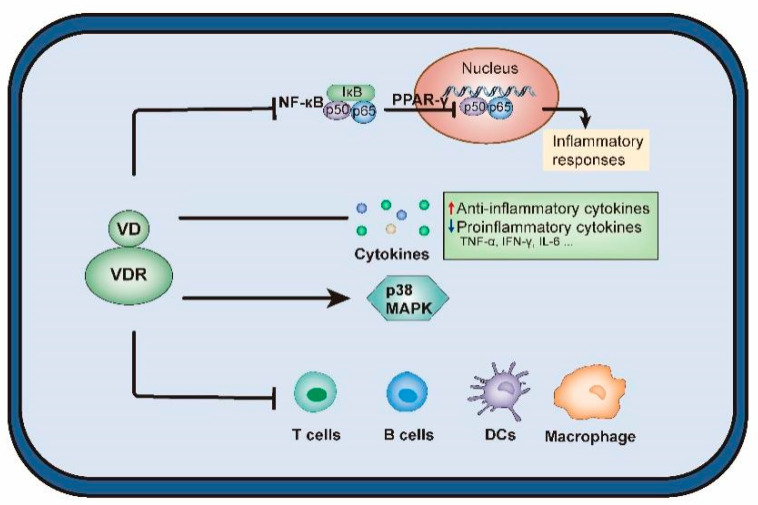
The effects of vitamin D on regulation of cytokine pathways. Abbreviations are as follows: VD, vitamin D; VDR, VD receptor; NF-κB, nuclear factor kappa-B; PPARγ, peroxisome proliferator-activated receptor γ; TNF-α, tumor necrosis factor-α; IFN-γ: interferon-γ, IL, interleukin; MAPK, mitogen activated protein kinases; DCs, dendritic cells.

**Table 1 nutrients-14-04230-t001:** Vitamin D status of pregnant women in different counties.

Country	Survey Year(s)	*N*	Deficiency Definition and Incidence	Insufficiency Definition and Incidence	Gestation Age	Measurement Method
Malaysia [8]	2016–2018	535	<30 nmol/L 228 (42.6%)	30–50 nmol/L 263 (49.2%)	Third trimester	Serum 25OHD concentration
Indonesia [9]	2016	160	<50 nmol/L 5 (3.1%)	50–75 nmol/L 93 (58.1%)	Third trimester	Serum 25OHD concentration
Vietnam [10]	2010–2012	960	Deficiency < 37.5 nmol/L Insufficiency 37.5–75 nmol/L 582 (60%) < 75 nmol/L		Third trimester	Serum 25OHD concentration
Thailand [11]	2011–2012	147	<50 nmol/L 50 (34.0%)	50–75 nmol/L 61 (41.5%)	Third trimester (at delivery)	Plasma 25OHD concentration
China [12]	2009	3598	<50 nmol/L First trimester 519 (37.15%) Second trimester 878 (62.85%)	50–75 nmol/L First trimester 444 (35.24%) Second trimester816 (64.76%)	First trimester Second trimester	Serum 25OHD concentration
Turkey [13]	2008	258	<50 nmol/L 233 (90.3%)	NA	Third trimester	Serum 25OHD concentration
India [14]	2006–2007	541	<50 nmol/L 521 (96.3%)	NA	All gestation age	Serum 25OHD concentration
Iran [15]	2002	552	<35 nmol/L 369 (66.8%)	NA	Third trimester (at delivery)	Serum 25OHD concentration
US [16]	2001–2006	841	<50 nmol/L First trimester 91 (46%) Second trimester 106 (32%) Third trimester 56 (18%)	50–75 nmol/L First trimester 73 (37%) Second trimester 142 (43%) Third trimester 91 (29%)	First trimester Second trimester Third trimester	Serum 25OHDconcentration
Sweden [17]	2008–2011	95	<30 nmol/L 16 (17%)	30–50 nmol/L 46 (48%)	Third trimester	Serum 25OHD concentration
Greece [18]	2003–2004	123	<25 nmol/L 24 (19.5%)	NA	Third trimester	Serum 25OHD concentration
Australia [19]	2003–2004	971	≤25 nmol/L 144 (15%)	26–50 nmol/L 317 (33%)	Third trimester	Serum 25OHD concentration
Brazil [20]	NA	190	<50 nmol/L First trimester 17 (23%) Second trimester 10 (9%)	50–75 nmol/L First trimester 32 (43%) Second trimester 47 (41%)	First trimester Second trimester	Serum 25OHD concentration
Kenya [21]	2011–2012	63	<50 nmol/L 13 (20.6%)	50–75 nmol/L 19 (30.2%)	Second trimester	Plasma 25OHD concentration

Here, 25OHD is 25-hydroxyvitamin D; the Institute of Medicine (IOM) defined a serum 25OHD level < 30 nmol/L as deficiency and 30–50 nmol/L as insufficiency; the International Osteoporosis Foundation (IOF) set a higher cutoff value for VD deficiency (25OHD < 50 nmol/L) and insufficiency (25OHD 50–75 nmol/L); NA indicates that data is not available.

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
