# Peer review of "Relationship between Maternal Vitamin D Levels and Adverse Outcomes"

_nutrients, 2022, doi:10.3390/nu14204230_

Round 1
Reviewer 1 Report
The article submitted for review comprehensively summarizes the knowledge about the relationship between the concentration of vitamin D in mothers and the course of pregnancy.
Remarks:
- in my opinion, the authors use incorrect structural formula of calcitriol throughout the text. Instead of 1,25-(OH)2D3, it should be 1,25(OH)2D because calcitriol is formed from 25(OH)D and not just only from 25(OH)D3 (lines 39, 130, etc.)
- please explain in more detail what it means "Approximately 10% of VD is obtained from dietary sources…" (line 44). Is it about 10% of vitamin D circulating in the blood or stored in the body ??? - it is not clear, although I know that such sentence appears very often in publications. However, this is not an argument and it needs to be clarified.
- in two places, the authors provide different criteria for vitamin D deficiency according to the Institute of Medicine. In lines 47-50 is "The Institute of Medicine (IOM) defined VD deficiency as 25OHD concentrations <20 ng/mL (50 nmol/L), VD insufficiency as 20 to 29 ng/mL (50-75 nmol/L), and VD adequacy as ≥ 30 ng/mL (75 nmol/L) in the serum [5] ", while in Table 1 is “the Institute of Medicine defined a serum 25OHD level <30 nmol/L as deficiency and 30- 50 nmol/L as insufficiency” (lines 64-65). This is completely incomprehensible and needs to be corrected.
- the sentence ”The need for VD increases during pregnancy owing to fetal skeletal growth needs, inadequate VD intake, and limited sunlight exposure” is not understandable (lines 52-53). According to the standards of many countries, including the opinion of the European Food Safety Authority, the need for vitamin D during pregnancy is the same as for non-pregnant women (15 µg). Moreover, it is not about the fact that the demand is increasing because the consumption of vitamin D is low, only that the supply of vitamin D in this period should covers the demand, due to the role of this vitamin in the course of pregnancy.
- why the consumption of this vitamin from the diet and supplementation are not mentioned among the factors determining the concentration of vitamin D "Many factors affect the VD status of 60 pregnant women, including latitude, season, time spent outdoors, clothing habits, sun-screen use, weight status , skin color, medications, and medical conditions ”(lines 60-62).
- there are no European countries in Table 1?
- whether in the face of ambiguous research results (as the authors themselves write in the summary of the article), instead of "Low maternal VD levels during pregnancy are associated with various adverse obstetric outcomes ..." (lines 68-69), should it not be "can be associated with various adverse obstetric outcomes… ”.
- what does it mean "dietary sources → vitamin D2" (figure 1). Vitamin D2 also comes from the diet. Such information may also suggest that vitamin D consumed with the diet, including vitamin D3, is transformed into vitamin D2 ???
- what does “1.25 DHCC/calcitriol” mean in this figure?
- throughout the article, I propose to write the concentration of vitamin D in the same units, either nmol/L, or ng/ml, or possibly both (one in parentheses). Now, in the sentences next to each other (line 101 and line 105), different units are given, which makes the article difficult to understand
- in line 136 - there are no vitamin D concentration units at all
Technical note - in many places the dot after the brackets should be removed (eg [52]. - line 163; [61]. - line 184; [64]. - line 200 etc.
Author Response
请参阅附件。

Reviewer 2 Report
This manuscript by Zhang et al gives an overview of the current research on Vitamin D and pregnancy. First is an overview of the evidence for associations between low vitamin D and a range of adverse pregnancy outcomes such as preterm birth, gestational diabetes and hypertension. The second half of the manuscript focuses on the functions of Vitamin D in pregnancy and how vitamin D deficiency may impact on these functions leading to various adverse outcomes. Overall, this manuscript was very clear and easy to read. There was a logical flow to the ideas and the sections were well constructed. Figure 1 very nicely reflected the ideas in the manuscript.
Specific comments:
1) Sections 3.1 and 3.2 are quite dense. It maybe be useful to add a pathway diagram to each of these sections to clearly indicate where in the processes Vit D is acting and what may be the outcome of low Vit D in terms of these signaling cascades.
2) line 339 ‘This highlights the essential role’
The evidence given in lines 335-338 do not indicate an essential role, but suggest one so I would suggest rephrasing line 339 to ‘This suggests an important role....’
3) line 347
It would be useful to definite what ‘Tregs’ are at this first mention.
4) line 204 ‘with adverse pregnancy outcomes’ is rather vague and perhaps the specific outcomes that were investigated could be used instead. Or perhaps use ‘with all adverse outcomes’ if more than stillbirth/abortion were investigated in this study.
5) section 2.1
One thing that seemed to be missing in the GDM section is an indication of the % of people suffering GDM that also have Vit D deficency. Is Vit D deficency more common in pregnant people who have GDM than in pregnant people who do not develop GDM?
6) line 94 ‘VD insufficiency leads to poor glucose tolerance, which causes the development of GDM’
Poor glucose tolerance is a defining feature of GDM not the underlying cause so this sentence should be rephrased.
Author Response
请参阅附件。
